# Long-Term Physical Activity Mitigates Inflammaging Progression in Older Adults Amidst the COVID-19 Pandemic

**DOI:** 10.3390/ijerph21111425

**Published:** 2024-10-27

**Authors:** Carlos André Freitas dos Santos, Ariane Nardy, Renato Jimenez Gomes, Brenda Rodrigues Silva, Fernanda Rodrigues Monteiro, Marcelo Rossi, Jônatas Bussador do Amaral, Vitória Paixão, Mauro Walter Vaisberg, Gislene Rocha Amirato, Rodolfo P. Vieira, Juliana de Melo Batista dos Santos, Guilherme Eustaquio Furtado, Ana Paula Ribeiro, Patrícia Colombo-Souza, Alessandro Ferrari Jacinto, Andre Luis Lacerda Bachi

**Affiliations:** 1Postgraduate Program in Translational Medicine, Department of Medicine, Paulista School of Medicine, Federal University of São Paulo (UNIFESP), São Paulo 04039-002, Brazil; afjacinto@unifesp.br; 2Postgraduate Program in Health Science, Santo Amaro University (UNISA), São Paulo 04829-300, Brazil; arianenardy@estudante.unisa.br (A.N.); rjimenez@estudante.unisa.br (R.J.G.); brendarodrigues@estudante.unisa.br (B.R.S.); monteiro.fernanda13@gmail.com (F.R.M.); mrossi.biotec@gmail.com (M.R.); anapribeiro@prof.unisa.br (A.P.R.); pcolombo@prof.unisa.br (P.C.-S.); albachi@prof.unisa.br (A.L.L.B.); 3ENT Research Laboratory, Department of Otorhinolaryngology—Head and Neck Surgery, Federal University of Sao Paulo (UNIFESP), São Paulo 04021-001, Brazil; jbamaral@unifesp.br (J.B.d.A.); v.paixao@unifesp.br (V.P.); vaisberg.mauro@gmail.com (M.W.V.); 4Mane Garrincha Sports Education Center, Sports Department of the Municipality of Sao Paulo (SEME), São Paulo 04039-034, Brazil; prof.gislene@hotmail.com; 5Post-Graduation Program in Science of Human and Rehabilitation, Federal University of São Paulo (UNIFESP), Santos 11010-150, Brazil; rodolfo.vieira@unievangelica.edu.br; 6Laboratory of Pulmonary and Exercise Immunology (LABPEI), Evangelical University of Goiás (UniEvangelica), Anápolis 75083-515, Brazil; 7Department of Physical Therapy, School of Medicine, University of São Paulo, São Paulo 05360-000, Brazil; juliana.mbs@usp.br; 8Polytechnic Institute of Coimbra, Applied Research Institute, 3045-093 Coimbra, Portugal; guilherme.furtado@ipc.pt; 9SPRINT Sport Physical Activity and Health Research & Innovation Center, Polytechnic University of Coimbra, 3045-601 Coimbra, Portugal; 10Center for Studies on Natural Resources, Environment, and Society (CERNAS), Polytechnic Institute of Coimbra, 3045-601 Coimbra, Portugal

**Keywords:** aging, older people, cytokines, physical exercise, gait speed, TUGT, handgrip, inflammaging

## Abstract

Background: Inflammaging and physical performance were investigated in older adults before and after the COVID-19 pandemic. Methods: Older women (*n* = 18) and men (*n* = 7) (mean age = 73.8 ± 7.1) were evaluated before the COVID-19 pandemic (PRE), 12 months after the lockdown (POST), and 10 months after resuming exercise training (POST-TR). Physical tests [gait speed (GS) and timed-up-and-go (TUG)]; muscle strength (handgrip—HG); and serum cytokine levels were assessed. Results: Older women showed higher GS and TUG at POST than PRE and POST-TR but lower HG at POST-TR than PRE, whereas older men exhibited lower HG at POST and POST-TR than PRE. Both groups presented (1) lower IL-10 and IL-12p70 values in contrast to higher IL-6/IL-10 and IL-8/IL-10 ratios at POST than PRE; (2) higher IL-10 values and lower IL-8/IL-10 ratio at POST-TR than POST; (3) higher IL-12p70/IL-10 ratio at POST-TR than PRE and POST. Particularly, older women showed (4) lower IL-6 values at POST and POST-TR than PRE; (5) lower IL-8 and IL-10 values at POST-TR than POST; (6) and higher TNF-α/IL-10 and IFN-γ/IL-10 ratios at POST than PRE and POST-TR. Significant correlations between the variables were found in both groups. Discussion: During the COVID-19 pandemic, detraining and resumption of exercise training promoted distinct alterations in physical capacity and inflammaging among older women and older men.

## 1. Introduction

Among the most remarkable achievements by humanity over the last two centuries is the increase in human lifespan, which was extended mainly due to the improvement of worldwide public health policies and modern medicine [1,2]. As a consequence of this triumph, the population over 65 years, which was around 8.5–9% in 2019, in accordance with the global estimate will increase to 12% in 2030 and 17% in 2050 [3]. Although the recent COVID-19 pandemic, unfortunately, caused a higher number of deaths, particularly of older adults, the estimates concerning the pace of population aging is still accelerated [4]. However, it is paramount to mention that living longer is not closely associated with living with health since it is evidenced that the expansion in lifespan is accompanied by the high prevalence of several chronic diseases and comorbidities, such as cancer, diabetes, cardiovascular, renal, lung, liver, neurological diseases, sarcopenia, frailty, and also an increased susceptibility to infectious diseases, which directly impacts the survival and well-being of older people [5,6]. Therefore, in order to minimize the burden on health systems and maintain quality of life in the aging process, the World Health Organization (WHO) has worked to stimulate public policies aimed at promoting healthy aging [7].

Regarding aging, it is a natural, dynamic, and multifactorial process characterized by a progressive decline of the majority of physiological systems [2,8]. Among these systems, the immunological is one of the most affected by aging and leads to the development of the phenomenon named immunosenescence, which affects both innate and adaptive immunity [9,10,11]. Since 2000, when Prof. Francheschi and collaborators published the article entitled “Inflammaging: an evolutionary perspective on immunosenescence” [12], the literature has highlighted that one of the main players involved in the development and progression of immunosenescence is the phenomenon of inflammaging, which is a chronic, sterile, low-grade inflammation associated with aging. It has been reported that the increased systemic levels of some pro-inflammatory mediators, particularly the C-reactive protein (CRP); interleukins (ILs) such as IL-1, IL-6, IL-8, IL-12; the tumor necrosis factor-alpha (TNF-α); and the interferon-gamma (IFN-γ), in association with the decreased levels of IL-1Ra and IL-10, two well-known anti-inflammatory cytokines, characterizes inflammaging [13,14]. Furthermore, it is worth mentioning that both these phenomena (immunosenescence and inflammaging) present a virtuous circle since one can fuel the other [15] and also can be pillars to the increased risk of the occurrence of chronic and infectious diseases in older adults [16].

In order to mitigate the deleterious effect of aging, especially in the older adult population, the consensus is that some approaches related to achieving an adequate nutritional status and regular physical exercise practice are considered powerful strategies in this context [17,18]. It is broadly accepted that the adoption and maintenance of an active lifestyle, through lifelong training, can maintain performance related to endurance and strength, as also muscle mass, favoring healthy aging in different ways, which includes the regulation of systemic inflammatory status by increasing the IL-10 and decreasing IL-6, IL-8, TNF-α, and CRP [15,19,20,21,22].

In contrast to the benefits of regular physical exercises in aging, a sedentary lifestyle and a decrease in strength and skeletal muscle are related to an increased risk of developing chronic diseases, comorbidities, and premature mortality [23,24]. In fact, studies have shown that a few weeks of interruption in physical exercise may be associated with reduced performance in functional physical tests and an increased risk of falls [25,26]. At this point, it is of utmost importance to highlight that, regarding the literature, during the confinement imposed by COVID-19, there was a significant elevation in sedentary behavior [27,28].

Although the WHO declared the end of the COVID-19 pandemic in May 2023, until now, the lockdown effects on the benefits of regular physical exercise performed by older adults in terms of systemic inflammatory status were scarcely assessed [29,30,31] and thus are not fully understood. Likewise, there is a lack of information on whether resuming exercise training could impact the inflammaging state in these same physically active older women and older men. Therefore, in the present study, we investigated the effect of the interruption of the regular physical exercise imposed by the COVID-19 pandemic (detraining period) and the consequences of the return to this exercise training program on inflammaging in a group of older adults.

## 2. Material and Methods

### 2.1. Study Design

This observational, retrospective, and prospective study, with blind analysis of outcomes and convenience sample, followed the Strengthening the Reporting of Observational Studies in Epidemiology (STROBE) guidelines to ensure methodological rigor and transparent reporting [32]. Anthropometric data, physical function tests, muscle strength, serum dosage of pro and anti-inflammatory cytokines, and their ratios in a group of older adults were assessed.

### 2.2. Participants Recruitment and Settings

This study involved community-dwelling older adults who were robust [33], physically very active [34], at no risk of malnutrition (evaluated by Mini Nutritional Assessment—MNA) [35], in agreement with a previous study [36], practiced moderate-intensity physical exercise for at least five years before the start of the COVID-19 pandemic, and resumed the same physical exercise program as lockdown measures were lifted and social isolation was eased. The participants were recruited and evaluated from the Primary Health Care Program for Older Adults belonging to Geriatric Discipline of the Federal University of São Paulo, Brazil, who practiced supervised physical exercises at the Municipal Education and Sports Center named “Mané Garrincha”, located in São Paulo City, Brazil. In 2019, a group of older volunteers were enrolled to participate in another study [36] which aimed to recognize the immune/inflammatory responses associated with the Influenza virus vaccination, as well as aspects of physical function associated with inflammaging in the aging process. At this moment, the pre-pandemic clinical and laboratory assessment was ratified (PRE, April–May 2019). However, with the declaration of the COVID-19 pandemic in 2020, we had to readapt our study due to the lockdown imposed in our city. Then, in 2021, after 12 months of the COVID-19 pandemic, we invited the volunteers to participate in a second assessment (POST, February–March 2021) before the volunteers received the first dose of the vaccine against the SARS-CoV-2 virus. In 2022, after permission from the local government to resume regular supervised physical exercise for 10 months, the third clinical evaluation was carried out (POST-TR, November–December 2022). Finally, in the months of January and February 2023, all older people who participated in any of the phases received feedback via telephone call on the results of their assessments.

### 2.3. Selection Criteria of Participants

All volunteers eligible were invited for participation in the study. Inclusion and exclusion criteria were carefully designed and evaluated by a single physician to ensure the safety and relevance of the participant pool. The inclusion criteria were as follows: (i) to be older than 60 years, (ii) have the autonomy to move to the exercise training center, (iii) receive medical clearance to engage in the exercise program, (iv) perform the same exercise program and be supervised by the same physical education professional, and (v) participate in the three clinical and laboratory evaluations of this study. The exclusion criteria were as follows: (i) present diagnosis of mental disease (dementia syndrome, major depression, severe mood disorder), neoplasm, renal and liver diseases, Type-1 Diabetes Mellitus, chronic infections, or out-of-control cardiovascular and metabolic disease; (ii) present symptoms of respiratory infections; (iii) infection by the SARS-CoV-2 virus during the study period; (iv) use of anti-inflammatory drugs or multivitamin/antioxidant supplements until the last 2 months before the evaluations. These criteria were essential to ensure the homogeneity of the study population and minimize confounding factors that could affect the outcomes of interest.

### 2.4. Sample Size Calculation

Based on the data concerning immune/inflammatory and physical function aspects presented in a previous study [37], the sample size and statistical power were estimated using an ANOVA test with the following parameters: effect size (*f*): 0.30; alpha error probability (*α*): 0.05; power (1-β error probability): 0.85; correlation among repeated measures: 0.5; and non-sphericity correction (*ε*): 1. Taken into account these inputs, a minimum of 35 older adults were required, with a margin of 30% for losses or refusal. This calculation was performed using the G*Power software (version 3.1.9.6) [38].

### 2.5. Ethical Procedures

All study procedures were thoroughly explained to the participants, who then provided informed consent, previously reviewed and approved by the local Research Ethics Committees (numbers: 3,623,247; 5,036,504; 4,350,476; 5,318,499) and by the National Research Ethics Committee (CAEE: 48166721.0.0000.5505), ensuring compliance with Brazilian Resolution (196/96) on research ethics with human subjects [39] and the Helsinki Declaration [40]. This study also followed international guidelines for ethics in physical exercise science research [41].

### 2.6. Outcomes Measures

#### 2.6.1. Combined-Exercise Training Program

The combined-exercise training program formerly was composed of aerobic and resistance exercises and was performed three times per week, lasting 60–75 min, for, at least, 24 months pre-pandemic and for 10 months in 2022 during the COVID-19 pandemic, following protocols described in the literature [9].

#### 2.6.2. Anthropometric Data and Physical Tests

The Body Mass Index (BMI) of each volunteer was calculated by measuring height and body weight on a digital scale and stadiometer (Personal^®^ scale, Filizzola, São Paulo, Brazil). Physical tests were assessed through the following tests: (i) time-up-go (TUG) in three meters on the way there and the same distance on the way back, and the results were expressed in seconds; (ii) gait speed (GS), on a four-meter route, starting at rest and disregarding the deceleration at the end of the route. The result was expressed in meters per second; and (iii) the muscle strength was evaluated through handgrip strength (HG) using an analog dynamometer (Jamar Hydraulic Hand Dynamometer^®^, Sammons Preston Rolyan, Bollingbrook, IL, USA), on which the result was shown in kilograms force. We used protocols previously described in our study [42].

#### 2.6.3. Blood Sample Collection and Analysis

Fasting-blood samples were collected between 8 and 9 h a.m. in tubes without an anticoagulant compound in order to obtain sera aliquots, on three different occasions: pre-pandemic (PRE, in 2019), after physical training stopped (POST, in 2021), and post-training-resumption (POST-TR, in 2022). Briefly, after blood coagulation, the tube was submitted to centrifugation (2000 rpm, 4 °C, 10 min), and a minimum of 500 mL of serum was added in Eppendorf’s tubes that were stored at −80 °C until the cytokines analyses. The volunteers were instructed not to perform physical activities of moderate or vigorous intensity in the 24 h prior to the collection.

Serum levels of pro- and anti-inflammatory cytokines (IL-6, IL-8, IL-10, IL-12p70, TNF-α, and IFN-γ) were determined using a multiplex assay (LEGENDplex™ bead-based multiplex assays, Biolegend, San Diego, CA, USA) and analyzed with a BD Accuri™C6 Plus Flow Cytometer (BD Biosciences San Jose, CA, USA), following the data analysis with LEGENDPlex™ V8.0 software (Biolegend). The concentration of these cytokines was calculated using appropriate standard curves (following instructions from manufacturers). The linearity of multiplex analysis of all cytokines assessed here was, respectively, within the range of 0–10,000 pg/mL, which includes the range of sample determinations. All correlation coefficients of standard curves were in the range of 0.93 to 0.99, whereas intra-assay coefficients of variance were 2–4% and inter-assay coefficients of variance were 7–10%. We also calculated the ratio between pro- and anti-inflammatory cytokines in order to evaluate the systemic inflammatory status [43].

#### 2.6.4. Statistical Analysis

Data normality and variance homogeneity were assessed using Shapiro–Wilk and Levene’s tests. Differences between parametric (anthropometrics and physical functional test) and non-parametric variables (cytokines) were analyzed using repeated-measures ANOVA with a Tukey’s post hoc test and a Friedman test with a Bonferroni’s post hoc test, respectively. Spearman´s rank correlation coefficient evaluated associations between variables. Additionally, effect size (ES) was calculated using Cohen’s coefficient. In this sense, values between 0.2 and 0.49 indicated a small effect, whereas between 0.5 and 0.79 indicated a moderate effect, and higher than 0.8 indicated a large effect. A significance level of 5% (*p* < 0.05) was set. Analyses were performed using GraphPad Prism version 10.1.1 (GraphPad Software, Boston, MA, USA).

## 3. Results

As shown in Figure 1, the total number of volunteers enrolled in this study was 25, lower than the value of the sample calculation, since among the 90 older people eligible for the study in 2019 (PRE), only 25 completed the two subsequent assessments (POST and POST-TR). In order to clarify this point, in the period between 2019 and 2022, 53 women and 12 men were excluded for one of the following reasons: refusal to be evaluated, changing city address, having had COVID-19, being unable to be located or not having returned to supervised physical training. Additionally, it is important to mention that no volunteer reported or even presented manifestations of anxiety/depression or even alteration in their cognitive level, as well as clinical manifestations of severe COVID and post-COVID syndrome during the study period.

Table 1 shows the data concerning anthropometric (age, weight, height, and BMI), functional physical tests (GS, TUG), muscle strength (HG), and clinical conditions of the older groups (women and men) participants in this study. Significant differences in chronological aging in both volunteer groups were found. In addition, the older women group presented higher GS and TUG values at POST than at PRE and lower HG values at POST-TR than at PRE. However, the same group showed lower GS values at POST-TR than at POST. Concerning the older men group, a significant decrease in HG values at POST and POST-TR as compared to PRE was found.

As presented in Figure 2, lower serum levels of IL-10 (Figure 2C) older women—ES = 0.62; older men—ES = 1.16) and IL-12p70 (Figure 2D, older women—ES = 0.44; older men—ES = 0.97) were found in both groups at POST than at PRE. A significant reduction in IL-6 (Figure 2A) was verified both at POST (ES = 0.32) and POST-TR (ES = 0.30) compared to PRE, particularly in older women. Additionally, IL-10 (Figure 2C) levels were significantly higher in both groups at POST-TR than at POST (older women—ES = 0.54; older men—ES = 0.71). Lastly, reduced levels of IL-8 (Figure 2B, ES = 0.45) and IL-10 (Figure 2C, ES = 0.30) were found in older women at POST-TR compared to PRE. Levels of IFN-γ (2E) and TNF-α (Figure 2F) were unchanged.

Figure 3 shows the ratios between pro-inflammatory cytokines and IL-10. At POST, whereas both groups presented a significant increase in the ratios of IL-6/IL-10 (Figure 3A, older women—ES = 0.52; older men—ES = 1.49) and IL-8/IL-10 (Figure 3B, older women—ES = 1.41; older men—ES = 1.38), a higher TNF-α/IL-10 ratio (Figure 3E, ES = 0.76) was observed in the older women than at PRE. In contrast, at POST-TR, not only the older women presented lower ratios of Il-6/IL-10 (Figure 3A, ES = 0.58), IFN-γ/IL-10 (Figure 3D, ES = 0.69), and TNF-α/IL-10 (Figure 3E, ES = 0.77), but also both groups showed a reduced IL-8/IL-10 ratio (Figure 3B, older women—ES = 1.06; older men—ES = 1.34) compared to POST. The IL-12p70/IL-10 ratio (Figure 3C) was higher in both groups at POST-TR than at PRE (older women—ES = 0.62; older men—ES = 0.77) and POST (older women—ES = 0.64; older men—ES = 0.71). Appendix A shows all values of serum cytokine levels and the ratios.

Data obtained in Spearman’s correlation analysis, particularly in the older women’s group (Table 2), revealed positive associations between BMI and TUG, and also between IL-6, IL-10, or IL-12p70 in all timepoints assessed. Age correlated positively with the IL-12p70/IL-10 ratio at PRE and POST-TR, whilst GS correlated positively with IL-10 at PRE. Several other positive correlations between anthropometrics, physical function tests, and cytokines were evidenced at PRE, POST, and POST-TR, whereas negative correlations were observed between GS and TUG at POS and POST-TR; GS and Age or IL-12p70/IL-10 ratio at PRE; and GS and IL-8/IL-10 ratio at POST. And exclusively at POST, the muscle strength, assessed by HG, and IL-12p70, or the ratios IL-12p70/IL-10 and IL-8/IL-10, were found.

Concerning the older men group, as shown in Table 3, positive correlations between IL-10 and IL-6 or IL-8 and a negative correlation between TUG and IL-10 were found at PRE- and POST-TR. The GS and IFN-γ were negatively correlated in PRE. The HG negatively correlated with the TNF-α/IL-10 ratio at POST and with TNF-α, IFN-γ, and IFN-γ/IL-10 ratio at POST-TR, in contrast to positive correlation with IL-10 at POST-TR. Other positive and negative correlations were found between anthropometrics, physical function tests, and cytokines on all occasions assessed here. Appendix A shows the values of cytokines and the ratios between IL-10 and the inflammatory cytokines.

## 4. Discussion

In general, our findings showed that (i) the interruption of regular physical exercise for one year negatively impacts the performance of physical tests and the control of the systemic inflammatory state; (ii) with the resumption of regular exercise, the reversal or attenuation of inflammaging was differently verified in the volunteer groups. In both cases, alterations in this phenomenon were more pronounced in the ratios analysis between pro- and anti-inflammatory cytokines, since this analysis allowed us to observe the regulated systemic inflammatory state found during the regular practice of physical exercise was significantly altered towards the pro-inflammatory state during the period in which regular practice of physical exercise was interrupted. Furthermore, even though both groups did not recover their muscle strength at the POST-TR time point, the stronger older men and those with better ability to realize in the mobility test showed better control of inflammaging after resuming physical exercise, whilst, in the older women group, the simple fact of returning to the regular practice of physical exercise mitigated or reversed this phenomenon.

Although we were able to present interesting findings, some limitations of the study are (a) the number of volunteers and the convenience sample to perform this study, mainly because of the refusal of many volunteers, initially invited, to continue in the study during the COVID-19 pandemic; (b) the discrepant number of participants in each volunteer group; (c) the absence of a comparison group comprising sedentary older individuals, encompassing both women and men; (d) more accurate measurements of physical exercise levels; (e) a lack of data from other physical function tests; (f) the absence of data collected immediately upon the resumption of supervised physical training; and (g) insufficient information about nutritional habits. Regarding this last item, it is important to mention that even though a nutritional assessment was carried out in 2019 (MNA) and no volunteer presented risk of malnutrition, this assessment is particularly qualitative and not quantitative [35]. In fact, it is known that an adequate quantitative consumption of macronutrients, especially proteins, can mitigate the development and progression of both immunosenescence and inflammaging [44,45].

Despite these facts, it is important to mention that our main results were obtained by evaluating and comparing the variables of each outcome measured for each volunteer, on the three occasions evaluated here, in duplicate analysis; thus, this can mitigate these drawbacks. It is also important to highlight that lockdown and social distancing ruled out the possibility of applying other physical tests to assess strength, which would require time for familiarization to be carried out, as we had already shown in other studies [42,46].

Since the aging process of skeletal muscle, immune, and inflammation occurs differently in men and women [8,47], male and female participants here were assessed separately. Additionally, it is worth mentioning that the volunteers were robust [48] and very active pre-pandemic, based on the previous observation that they performed 600 min of moderate-intensity activities per week, which was assessed using the IPAQ questionnaire (International Physical Activity Questionnaire) [34], and also presented positive adaptations in physical and inflammatory aspects [46]. Interestingly, demographic studies have shown that around a quarter (25%) of adults (particularly over 50 years old) [49,50,51,52] and approximately 5% of the older adults’ population [53] perform moderate-intensity physical activities with a similar quantity to our volunteer group. Thus, even with the decrease in physical performance during the COVID-19 pandemic, the data found in the assessments of the physical characteristics, particularly BMI, HG, and TUG tests, suggest that volunteers maintained adequate functional ability [54].

It was reported that lockdown related to the COVID-19 pandemic increased sedentary behavior and detraining [27,28], which could putatively justify the worsened results of the physical tests found here. Additionally, some findings of this study can reinforce already-known associations, such as the influence of age and BMI in physical tests involving body movement, and the fact that GS and TUG tests can evaluate similar muscle functions [42]. In this sense, studies performed during the COVID-19 pandemic were able to reveal either the nullification of physical and metabolic benefits of exercise training after detraining [55] or the maintenance of physical performance, even after twenty-eight weeks of detraining imposed by the pandemic [56]. For instance, a reduction of 5%, 15%, and 17% in muscle strength and power and type II thigh muscle fibers after 12 weeks of social isolation, respectively [57], as well as a decrease in muscle functions, type II fibers, and satellite cells post-12 months of sedentary behavior, were reported [58]. As expected, our results showed differences between the groups, since after twelve months of detraining, the older women group showed reductions of 37% and 7%, in GS and TUG tests, respectively, and the older men group showed a decrease around of 10% in HG. Thus, specifically during this period, while the group of older women lost the fitness to perform functional physical tests of the lower limbs, the older men group decreased upper limb muscle strength. Some possible explanations for these discrepant observations could be associated with the fact that (i) in the aging process, the muscular architecture can be impacted by some factors, such as hormonal, genetic, inflammatory, behavioral, and nutritional factors [59,60], which can lead women to lose muscle strength earlier than men [61]. Taken these data into account, our findings showed that, unexpectedly, there was an absence of significant muscle strength loss in the women’s group, particularly during the detraining phase. Although we cannot affirm, based on the literature, a reasonable suggestion for this observation could be related to the fact that women can have better adaptation to the endurance of the muscles of the hands and wrists [62,63], which could be related to physical activities that involve domestic tasks [64]. Moreover, (ii) by generally being stronger [61], the ceiling effect [65] could have occurred in the application of the physical tests in the older men, as the GS and TUG have low correlation with classic strength tests [42].

Another remarkable observation, evidenced here, was that the return to exercise training did not “reverse” or mitigate the effect of detraining in the muscle strength in both groups. However, the older women group showed values of the GS test similar to the pre-pandemic phase, and, based on the literature, improvement in this parameter has been associated with positive clinical outcomes [66]. In fact, studies demonstrated that twelve weeks of regular physical training, after eight weeks of detraining, should be sufficient to recover the architecture, strength, and muscle power [57]. Moreover, in an intention to prevent a sedentary lifestyle during the COVID-19 pandemic, it was reported that ten weeks of physical exercise intervention favored the increment of muscle strength in adults with multiple sclerosis [67].

Likewise, notable neuromuscular adaptations could be achieved in these volunteers after ten months of regular physical training practice [68]. Despite the physical aspects of ‘muscle memory’ [69] not being evaluated here, the fact is that none of the volunteer groups, after 10 months of training, reached the strength measured by the HG of the pre-pandemic period, even if it could be expected [70]. Regarding muscle plasticity after a long period of detraining, it is known that epigenetic factors are fundamental [71]; so, the significant reduction in muscle strength assessed by the HG, an average of 10% in both groups, could be supposedly associated not only with the aging process but also with behavioral, dietary, and nutritional issues related to the pandemic, recalling that none of the volunteers presented clinical symptoms related to COVID-19, including severe symptoms and post-COVID-19 syndrome, during the study.

Each day, studies present different biomarkers that can be useful to determine both successful aging and vulnerabilities to frailty, as well as the physical performance and inflammaging phenomenon [2]. In this respect, it is known that cytokines present a myriad of actions, and older adults who regularly practice physical exercises presented an exemplary phenotype of the best regulations of inflammaging [19]. Although the systemic levels of IL-6, IL-8, IL-12p70, and IL-10 were higher at PRE, and these findings putatively suggested that long-standing physical exercise would drive the systemic state toward a pro-inflammatory direction, the data obtained in the ratios between pro- and anti-inflammatory cytokines, in general, demonstrated an inflammatory status regulated pre-pandemic [46], a significant deregulation during detraining phase [31,72], and a “return” of regulated status with the resumption of exercise training [19]. In fact, physical exercise is an activity that regulates inflammation in chronically active older people and in those who start exercising late [30].

Based on our results, these variations in inflammatory status found in both groups can closely be related to the anti-inflammatory properties of IL-10 [13], which were reduced in the detraining phase and increased with the resumption of exercise training. It is known that exercise training can improve circulating IL-10 levels, which strengthens its role in the regulation of inflammaging [19]. During physical exercise, muscle contraction activates the transcription of several genes, mainly IL-6 that, among other actions, promote IL-10 releases, which lead to the control of inflammation in this context [17]. Particularly, our data concerning the ratios between the pro-inflammatory cytokines IL-6, IL-8, TNF-α, and IFN-γ, with IL-10, reinforce this remarkable capacity of exercise training in improving IL-10 levels, which, consequently, can favor the regulation of the systemic inflammatory state, including in older populations. Likewise, the increase in the IL-12p70/IL-10 ratio in 2022 (at POST-TR) not only is interesting but also could represent a lack of systemic inflammatory control. However, the literature claims that IL-12p70 is pivotal both in increasing the immune cells’ activity, especially Natural Killer (NK) cells, and its increase has been associated with successful aging [73].

Taken together, our data allow us to suppose that the interruption of regular physical exercise during the COVID-19 pandemic, in this physically active older population, led to a prominent imbalance in systemic cytokine levels that favored the progression of the inflammatory phenomenon, as previously reported in sedentary older people and detrained individuals [20,74], also related to COVID-19 [31]. It is paramount to point out that this finding is very important due to the fact that it has been reported that inflammaging can make older people more susceptible to negative immunological outcomes, with remarkable impairment in both innate and acquired immune responses [12,75].

Regarding the correlations between physical function tests and cytokines, during the detraining period, it was found that better physical performance was associated with the best regulation of the systemic inflammatory state. Despite the fact that the older women group did not present significant changes in the application of force on HG, between the pre-pandemic and during the detraining period, the correlations found illustrated that the stronger and faster volunteer presented advantages in the systemic inflammatory profile. Specifically, for the older men group, the HG test was the most important evaluation both in the perception of strength loss during the detraining period and in the interpretation that greater muscle strength was associated with a balanced systemic inflammatory status. Indeed, the association between muscle strength and positive health outcomes has been the subject of investigation for a long time and seems to be a ‘two-way street’ [17]. Based on these aspects and our results, we can suggest that not only the lifestyle alterations associated with the COVID-19 pandemic could negatively influence the benefits of years of regular physical training practice, but also those physically fitter older adults might be less affected, as previously described in the literature [19,24].

In this sense, the associations between the fitness to realize the physical tests and the inflammatory phenomenon found after resuming training in the group of older men can reinforce the previous suggestion that stronger and faster volunteers might generate better conditions to mitigate the development and progression of inflammaging [24,30]. However, in the older women group, the improved GS test results after return to training could influence the ratios between pro-inflammatory (IL-6, IL-8, TNF-α) and anti-inflammatory (IL-10) cytokines to become similar to the pre-pandemic period. These results strengthen the differences between genders during aging, although the increase in BMI and age can supposedly affect this balance [46]. Therefore, even with the possible ceiling effect in mobility tests in the older men’s group, these differences between the volunteer groups during detraining and the resumption of supervised physical training, during the COVID-19 pandemic, are unprecedented and can corroborate the literature, as both in vitro and/or in vivo studies have shown that men and women are differently affected by pro-inflammatory cytokines [47,76].

It is worth mentioning that our findings reinforce the perception that multiple factors are involved in the inflammaging both in men and women. For instance, it has been reported that hormonal, genetic, and epigenetic factors could favor the generation of robust anti-inflammatory responses in women [47]. Furthermore, there is an understanding that the cytokine network is pleiotropic, as it presents redundant and overlapping activities associated with chemotaxis, proliferation, development, activation, and migration of different types of cells (immune, neuronal, muscular, and vascular endothelial, among others), in addition to demonstrating positive or negative actions, particularly related to their levels and chronicity [77]. Figure 4 is a summary of the main results of this study.

## 5. Conclusions

The findings of this study lead us to propose that (a) the regular practice of combined exercise training can benefit and help in regulating systemic inflammatory status; (b) the period of detraining significantly impacted physical functional capacity and contributed to the progression of the inflammaging process; and also, (c) older individuals, both women and men, were differently influenced by both the detraining and the resumption of exercise training.

## Figures and Tables

**Figure 1 ijerph-21-01425-f001:**
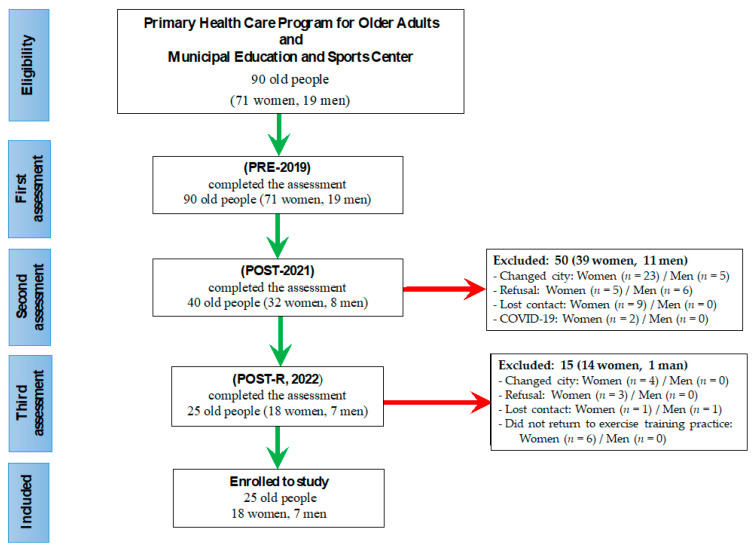
STROBE-flow diagram. The green arrows indicate the flow of the volunteer evaluation steps, and the red arrows indicate the flow of the volunteer exclusion steps.

**Figure 2 ijerph-21-01425-f002:**
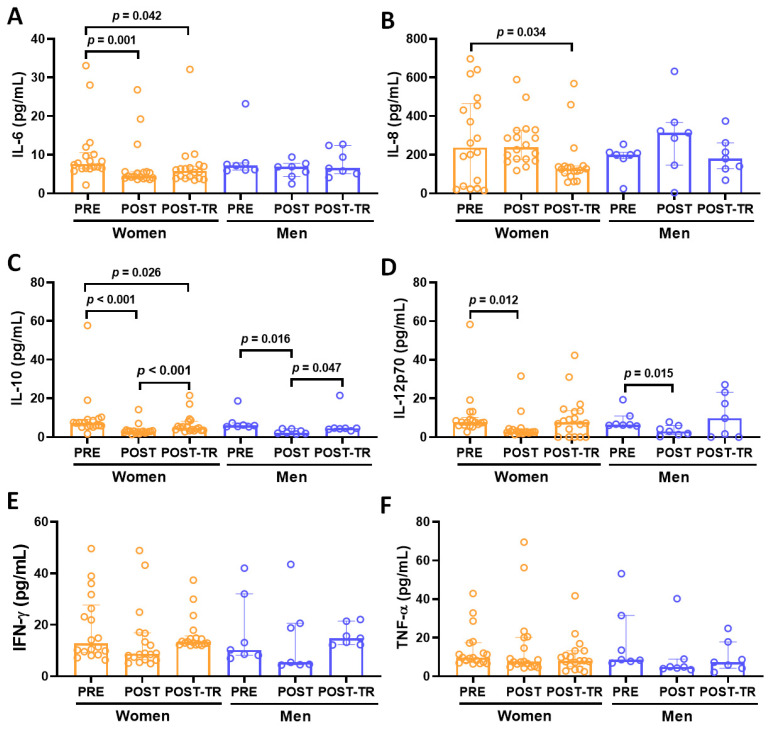
Serum cytokines levels, presented in median and interquartile range, of IL-6 (**A**), IL-8 (**B**), IL-10 (**C**), IL-12p70 (**D**), IFN-γ (**E**), and TNF-α (**F**) in the volunteer groups (older women and older men) before COVID-19 pandemic (PRE) after 12 months of interruption (POST, detraining phase) and also after 10 months of resuming of regular practice of combined exercise training program (POST-TR). The significant level was set as 5% (*p* < 0.05).

**Figure 3 ijerph-21-01425-f003:**
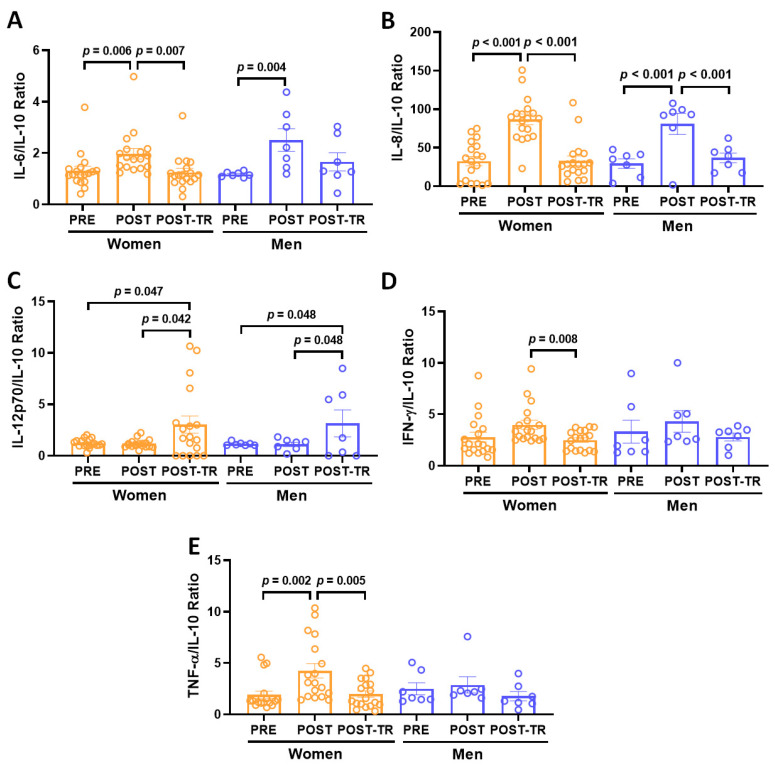
Ratio, in median and standard deviation, between IL-6/IL-10 (**A**), IL-8/IL-10 (**B**), IL-12p70/IL-10 (**C**), IFN-γ/IL-10 (**D**), and TNF-α/IL-10 (**E**) in the volunteer groups (older women and older men) before COVID-19 pandemic (PRE) after 12 months of interruption (POST, detraining phase) and also after 10 months of resuming of regular practice of combined exercise training program (POST-TR). The level of significance was established at 5% (*p* < 0.05).

**Figure 4 ijerph-21-01425-f004:**
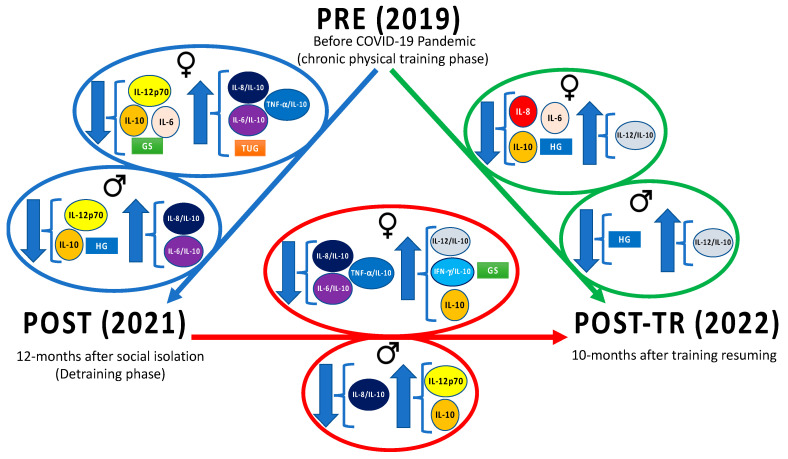
Summary of significant results concerning physical functional tests and inflammatory cytokines in volunteer groups, both older women and older men assessed on three different occasions: (1) before the COVID-19 pandemic (PRE), a period in which they performed a mean of 600 min per week of a moderate-intensity combined exercise training in moderate intensity; (2) after 12 months of interruption of regular practice of exercise training (POST, detraining phase); and (3) after 10 months of resumption of regular practice of combined exercise training program (POST-TR), in a similar way to the performed at PRE time point (mean of 600 min per week in moderate intensity). In the blue arrow are shown the significant results obtained in the comparison between PRE and POST time points. In the green arrow are shown the significant results obtained in the comparison between PRE and POST-TR time points. In the red arrow are shown the significant results obtained in the comparison between POST and POST-TR time points. GS—gait speed; HG—handgrip; IFN—interferon; IL—interleukin; TNF—tumor necrosis factor; TUG—timed-up-go.

**Table 1 ijerph-21-01425-t001:** Data concerning demographic and anthropometric characteristics, physical function tests (GS, TUG, and HG), all presented as mean and standard deviation, as well as clinical conditions (*n* = volunteers number) of the older women and older men groups in the different occasions: before COVID-19 pandemic (PRE), after 12 months of interruption (POST, detraining phase), and also after 10 months of resuming of regular practice of the combined-exercise training program (POST-TR). The level of significance was established at 5% (*p* < 0.05).

Variables	Volunteer Groups								
	Older Women (*n* = 18)			*p*-Values		Older Men (*n* = 7)			*p*-Values
	PRE	POST	POST-TR			PRE	POST	POST-TR	
Age (years)	75.2 ± 7.0 *	76.2 ± 7.0	77.1 ± 7.1 ^†^	*^,†^ <0.001		72.4 ± 7.1 *	73.4 ± 7.1	75.1 ± 7.1 ^†^	*^,†^ <0.001
Weight (kg)	60.32 ± 14.5	59.78 ± 14.23	59.58 ± 14.11	0.534		70.69 ± 9.14	67.61 ± 10.79	70.96 ± 8.71	0.312
Height (m)	1.53 ± 0.07	1.53 ± 0.07	1.53 ± 0.07	0.955		1.68 ± 0.08	1.68 ± 0.08	1.68 ± 0.08	0.915
BMI (kg/m^2^)	24.32 ± 4.21	24.14 ± 4.46	24.29 ± 4.16	0.694		24.82 ± 3.29	23.59 ± 2.05	25.30 ± 2.98	0.401
GS (m/s)	0.97 ± 0.21 *	0.81 ± 0.25	0.99 ± 0.18 ^†^	* 0.009		1.03 ± 0.15	1.06 ± 0.18	1.02 ± 0.19	0.914
				^†^ 0.015					
TUG (s)	6.8 ± 1.0 *	7.3 ± 1.2	7.1 ± 0.8	* 0.033		6.0 ± 0.5	6.1 ± 0.4	6.3 ± 0.6	0.707
HG (kgf)	23.1 ± 3.7 ^#^	21.3 ± 4.5	20.7 ± 3.6	^#^ 0.005		37.6 ± 5.9 *^,#^	33.6 ± 7.5	33.8 ± 7.4	* 0.008
									^#^ 0.019
Clinical conditions (*n*, all verified in 2019)									
Very.active/IPAQ	18				7				
Sarcopenia	0				0				
Malnutrition/MNA	0				0				
T2DM	1				0				
Dyslipidemia	1				1				
Hypertension	0				2				

Note: BMI, body mass index; GS, gait speed; TUG, timed-up-and-go; HG, handgrip; Very.active/IPAQ, very active physical activity level/International Physical Activity Questionnaire [34]; MNA, mini nutritional assessment [35]; T2DM, type-2 diabetes mellitus; * significant difference between the values obtained in PRE and POST; ^#^ significant difference between the values obtained in PRE and POS-TR; ^†^ significant difference between the values obtained in POST and POST-TR.

**Table 2 ijerph-21-01425-t002:** Significant results of the Spearman’s correlation coefficient analysis between age, anthropometric characteristics, the physical function tests, systemic cytokine levels, and the ratio between pro- and anti-inflammatory cytokines in the old women group, obtained in the different occasions: before COVID-19 pandemic (PRE), after 12 months of interruption (POST, detraining phase), and also after 10 months of resuming of regular practice of the combined-exercise training program (POST-TR). The level of significance was established at 5% (*p* < 0.05).

Older Women Group (*n* = 18)								
Parameters	2019 (PRE)		Parameters	2021 (POST)		Parameters	2022 (POST-TR)	
	Rho-Value	*p*-Value		Rho-Value	*p*-Value		Rho-Value	*p*-Value
BMI × TUG	0.60	0.009	BMI × TUG	0.65	0.009	BMI × TUG	0.53	0.025
Age × GS	−0.56	0.016	GS × TUG	−0.63	0.012	GS × TUG	−0.54	0.020
Age × IL-12p70/IL-10	0.58	0.012	GS × IL-8/IL-10	−0.51	0.044	BMI × TNF-α	0.48	0.045
GS × IL-12p70/IL-10	−0.57	0.014	TUG × IL-8	0.55	0.034	Age × IL-6	0.53	0.023
GS × IL-10	0.50	0.036	HG × IL-10	0.49	0.041	Age × IL-12p70	0.49	0.038
IL-6 × IL-10	0.99	<0.001	HG × IL-12p70	−0.51	0.030	Age × IL-12p70/IL-10	0.48	0.044
IL-6 × IL-12p70	0.99	<0.001	HG × IL-12p70/IL-10	−0.51	0.032	IL-6 × IL-10	0.77	<0.001
IL-6 × TNF-α	0.99	<0.001	HG × IL-8/IL-10	−0.56	0.016	IL-6 × IL-12p70	0.58	0.011
IL-6 × IFN-γ	0.91	<0.001	IL-6 × IL-10	0.81	<0.001	IL-10 × IL-12p70	0.70	0.001
IL-10 × IL-12p70	0.99	<0.001	IL-6 × IL-12p70	0.82	<0.001			
IL-10 × TNF-α	0.99	<0.001	IL-10 × IL-12p70	0.99	<0.001			
IL-10 × IFN-γ	0.91	<0.001	IL-8 × TNF-α	0.64	0.004			
IL-12p70 × TNF-α	0.99	<0.001	IL-8 × IFN-γ	0.59	0.011			
IL-12p70 × IFN-γ	0.90	<0.001	TNF-α × IFN-γ	0.86	<0.001			

Note: BMI, body mass index; GS, gait speed; TUG, timed-up-and-go; HG, handgrip; IL, interleukin; TNF-α, tumor necrosis factor-alpha; IFN-γ, the interferon-gamma.

**Table 3 ijerph-21-01425-t003:** Significant results of the Spearman’s correlation coefficient analysis between age, anthropometric characteristics, the physical function tests, systemic cytokine levels, and the ratio between pro- and anti-inflammatory cytokines in the old man group, obtained in the different occasions: before COVID-19 pandemic (PRE), after 12 months of interruption (POST, detraining phase), and also after 10 months of resuming of regular practice of combined exercise training program (POST-TR). The level of significance was established at 5% (*p* < 0.05).

Older Men Group (*n* = 7)								
Parameters	2019 (PRE)		Parameters	2021 (POST)		Parameters	2022 (POST-TR)	
	Rho-Value	*p*-Value		Rho-Value	*p*-Value		Rho-Value	*p*-Value
GS × IFN-γ	−0.78	0.049	Age × TUG	0.81	0.029	BMI × IL-6	0.76	0.049
TUG × IL-10	−0.79	0.048	HG × TNF-α/IL-10	−0.79	0.048	TUG × IL-10	−0.87	0.010
IL-6 × IL-10	0.96	0.003	IL-10 × IL-12p70	0.95	<0.001	TUG × IL-8	0.81	0.027
IL-6 × IL-12p70	0.86	0.024	IL-10 × IFN-γ	0.90	0.006	HG × TNF-α	−0.79	0.033
IL-8 × IL-10	0.79	0.048	IL-12p70 × IFN-γ	0.93	0.002	HG × IFN-γ	−0.83	0.022
IL-10 × TNF-α	0.86	0.024	TNF-α × IFN-γ	0.85	0.016	HG × IFN-γ/IL-10	−0.73	0.048
						IL-8 × IL-10	0.80	0.032
						IL-8 × TNF-α	0.77	0.041
						IL-10 × TNF-α	0.79	0.035
						TNF-α × IFN-γ	0.85	0.015

Note: BMI, body mass index; GS, gait speed; TUG, timed-up-and-go; HG, handgrip; IL, interleukin; TNF-α, tumor necrosis factor-alpha; IFN-γ, the interferon-gamma.

## Data Availability

The original contributions presented in the study are included in the article/Appendix A, further inquiries can be directed to the corresponding author/s.

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
