# Peer review of "Long-Term Physical Activity Mitigates Inflammaging Progression in Older Adults Amidst the COVID-19 Pandemic"

_ijerph, 2024, doi:10.3390/ijerph21111425_

Round 1
Reviewer 1 Report
Comments and Suggestions for Authors
Dear authors,
The article investigates how long-term physical activity can mitigate inflammation progression in the elderly during the COVID-19 pandemic. However, some methodological aspects need to be clarified and some results discussed in more detail. Following are some suggestions:
1. I believe the major limitation of the study is the small sample size (25 participants) and the lack of a control group, which may limit the generalizability of the results:
· The drop in the number of participants from the beginning (90) to the end is significant and introduces a potential selection bias, especially considering that those excluded might have different characteristics than the remaining participants.
· In addition, The lack of a control group consisting of sedentary subjects limits the possibility of comparing results and assessing whether the effect of training was truly significant compared with a population that did not resume any physical activity. This is a critical point in determining the true effect of resuming physical activity.
2. Line 321-322: Although the study distinguishes between elderly men and women, gender differences in inflammatory response and physical capacity deserve more discussion: It is mentioned that women lost leg strength faster, but potential causes should be developed further.
3. The study does not consider or report detailed information regarding participants: for example, diet, which could be a significant confounding factor in determining inflammation and response to exercise. In line 353 you mentioned the “cultural reasons” and the sentence “women were possibly involved with domestic tasks, which could supposedly attenuate the reduction of muscle strength in this group” However, this might turn out to be a little premature conclusion also given the elderly age of the subjects. Did you collect information about daily life?
4. In the conclusions session you state that “the regular practice of combined exercise training can benefit and help in regulating systemic inflammatory status” It would be interesting to briefly discuss the protocols deepening the FITT (frequency, intensity, time, and type) characteristics of training administered during COVID-19 to fragile populations to emphasize the practical application of the study e.g. Amato et al. demonstrates how 10 weeks of training during pandemic improves HG test in fragile subjects (people with MS)
Amato A, Messina G, Feka K, Genua D, Ragonese P, Kostrzewa-Nowak D, Fischetti F, Iovane A, Proia P. Taopatch® combined with home-based training protocol to prevent sedentary lifestyle and biochemical changes in MS patients during COVID-19 pandemic. Eur J Transl Myol. 2021 Aug 31;31(3):9877. doi: 10.4081/ejtm.2021.9877. PMID: 34498450; PMCID: PMC8495370.
Author Response
"Please see the attachment."

Reviewer 2 Report
Comments and Suggestions for Authors
The average age of the participants should be mentioned in the abstract.
P-values ​​for all comparisons should be mentioned in the abstract.
The introduction is well written. However, considering the large number of studies in recent years, it is suggested to mention more backgrounds in the introduction.
References
Modaberi S, Saemi E, Federolf PA, van Andel S. A systematic review on detraining effects after balance and fall prevention interventions. Journal of clinical medicine. 2021 Oct 11;10(20):4656.
Aragão-Santos JC, Pantoja-Cardoso A, Dos-Santos AC, Behm DG, de Moura TR, Da Silva-Grigoletto ME. Effects of twenty-eight months of detraining imposed by the COVID-19 pandemic on the functional fitness of older women experienced in concurrent and functional training. Archives of Gerontology and Geriatrics. 2023 Aug 1;111:105005.
What statistical method did you use to estimate the sample size with GPower software? please mention Also, cite a source for the effect size mentioned.
Why did you choose participants from both genders? Was the comparison between the two sexes one of your main goals? If it is, it should be mentioned in the title.
The results are well reported. Of course, it is suggested to report the graphs with distinct colors so that the groups are clear.
The discussion is also well written. However, as mentioned in the introduction section, it is suggested to discuss related and new researches.
Author Response
"Please see the attachment."

Reviewer 3 Report
Comments and Suggestions for Authors
I would like to express my most sincere thanks for the opportunity to review this interesting and thorough study. The work presented is a significant contribution to understanding the impact of the interruption and resumption of regular physical exercise on the elderly during the COVID-19 pandemic. The quality of the research is evident through the detailed methodology, accurate statistical analyses, and the importance of the results.
Strengths of the paper:
- The adoption of a well-defined and detailed study design, including assessments before the pandemic, post-exercise interruption, and post-exercise resumption, provides a comprehensive view of the impact of these changes on the elderly.
- The use of specific functional tests, such as Gait Speed (GS) and the "Timed Up and Go" (TUG), along with the measurement of pro- and anti-inflammatory cytokines, is a robust approach to evaluate physical and inflammatory status.
- The analysis of gender differences in inflammatory and physical outcomes adds value, considering the scarcity of studies on this specific topic.
Suggested modifications:
-
Line 32-33: Correct "POST-TR" to ensure consistency in term usage (minor correction).
-
Line 37: "lower IL-8 and IL-10 values" – Consider using a more precise terminology like "reduced concentrations of IL-8 and IL-10" for scientific clarity.
-
Line 109-111: Provide more details about the participant selection criteria, specifically clarifying whether those who experienced severe COVID-19 complications during the study were also excluded to avoid confusion.
-
Line 146: Specify the confidence interval used for sample size calculation, as only the statistical power (95%) is indicated.
-
Line 295-297: I suggest expanding the analysis of the "more pronounced alterations regarding the ratios between pro- and anti-inflammatory cytokines" by providing a more detailed explanation of the significance of these cytokine ratios.
-
Figure 1: In the figure legend, better clarify the interpretation of the arrows to indicate which groups were included and excluded at the various stages.
-
Limitations Section (Line 306): The authors should explicitly reference the missing nutritional data, emphasizing how this could affect systemic inflammation and physical performance outcomes.
Parts to eliminate or simplify:
-
Line 378-381: The sentence "Of note, the cytokine levels found in our older volunteers on the three occasions... were similar to those evidenced in a healthy adult population" seems redundant since cytokine data are already extensively discussed. I suggest removing or shortening this section.
-
Table 3 and detailed correlations: Some correlations, while statistically significant, do not add new insights to the discussion (e.g., GS x IFN-γ). I recommend keeping only the correlations that contribute meaningfully to the overall narrative.
Study limitations: Despite the strong conclusions, several limitations emerge:
- Small sample size (Line 306-310): The sample size is smaller than initially estimated, which may reduce the statistical power and generalizability of the results.
- Lack of nutritional data (Line 311-312): The absence of detailed nutritional information on participants may influence the results related to systemic inflammation responses.
- No control group (Line 309): The absence of a sedentary control group limits the ability to draw strong conclusions about the protective effect of physical exercise.
Missing citations: I recommend adding the following two references in the discussion to strengthen the conclusions regarding the importance of physical exercise during the COVID-19 isolation period:
-
D'Oliveira et al. (2022) to support the concept that home-based exercise was effective in improving both mental and physical health during the pandemic. This citation can be included in the discussion section related to the impact of exercise interruption on inflammation (after line 409).
-
Da Cruz et al. (2022), which explores the role of physical activity in social isolation among the elderly, can be inserted in the paragraph discussing strategies to mitigate the negative impact of sedentary behavior during the lockdown (after line 421).
full references:
D'Oliveira, A., De Souza, L. C., Langiano, E., Falese, L., Diotaiuti, P., Vilarino, G. T., & Andrade, A. (2022). Home Physical Exercise Protocol for Older Adults, Applied Remotely During the COVID-19 Pandemic: Protocol for Randomized and Controlled Trial. Frontiers in psychology, 13, 828495. https://doi.org/10.3389/fpsyg.2022.828495 da Cruz, W. M., D' Oliveira, A., Dominski, F. H., Diotaiuti, P., & Andrade, A. (2022). Mental health of older people in social isolation: the role of physical activity at home during the COVID-19 pandemic. Sport sciences for health, 18(2), 597–602. https://doi.org/10.1007/s11332-021-00825-9
Author Response
"Please see the attachment."

Round 2
Reviewer 1 Report
Comments and Suggestions for Authors
Dear Authors,
you have fully and completely responded to all my comments. In particular, they have enriched and clarified some information regarding participantes characteristics that was missing. Also, the justification you gave to the small sample size and the absence of the control group given the singularity of the sample and the retrospective nature of the study might be acceptable.
Reviewer 3 Report
Comments and Suggestions for Authors
I confirm that the required revisions to the manuscript have been successfully completed by the authors, following the provided guidelines. The document is now ready for publication.
I thank the authors for their cooperation and professionalism in responding to the directives and bringing the text to the required standards.
Please proceed with the final stages of publication.